# Effect of Nanoparticles Surface Charge on the *Arabidopsis thaliana* (L.) Roots Development and Their Movement into the Root Cells and Protoplasts

**DOI:** 10.3390/ijms20071650

**Published:** 2019-04-03

**Authors:** Anna Milewska-Hendel, Maciej Zubko, Danuta Stróż, Ewa U. Kurczyńska

**Affiliations:** 1Department of Cell Biology, Faculty of Biology and Environmental Protection, University of Silesia in Katowice, 28 Jagiellońska Street, 40-032 Katowice, Poland; 2Institute of Materials Science, Faculty of Computer Science and Materials Science, University of Silesia in Katowice, 75 Pułku Piechoty Street 1a, Chorzów, 41-500, Poland; maciej.zubko@us.edu.pl (M.Z.); danuta.stroz@us.edu.pl (D.S.); 3Department of Physics, University of Hradec Králové, Hradec Králové 500-03, Czech Republic

**Keywords:** *Arabidopsis thaliana*, gold nanoparticles, protoplasts, root development, root meristem, ultrastructure

## Abstract

Increasing usage of gold nanoparticles (AuNPs) in different industrial areas inevitably leads to their release into the environment. Thus, living organisms, including plants, may be exposed to a direct contact with nanoparticles (NPs). Despite the growing amount of research on this topic, our knowledge about NPs uptake by plants and their influence on different developmental processes is still insufficient. The first physical barrier for NPs penetration to the plant body is a cell wall which protects cytoplasm from external factors and environmental stresses. The absence of a cell wall may facilitate the internalization of various particles including NPs. Our studies have shown that AuNPs, independently of their surface charge, did not cross the cell wall of *Arabidopsis thaliana* (L.) roots. However, the research carried out with using light and transmission electron microscope revealed that AuNPs with different surface charge caused diverse changes in the root’s histology and ultrastructure. Therefore, we verified whether this is only the wall which protects cells against particles penetration and for this purpose we used protoplasts culture. It has been shown that plasma membrane (PM) is not a barrier for positively charged (+) AuNPs and negatively charged (−) AuNPs, which passage to the cell.

## 1. Introduction

Substantial progress in the development of nanotechnology in recent years has led to an increase in production and widespread usage of nanomaterials (NMs) in many application fields [1,2]. This remarkable progress is inevitably accompanied by the risk of NMs being released into the environment [3]. As a result, NPs enter an ecosystem and may cause an invisible danger to the environment by interaction with the living organisms [4,5,6]. Plants as an essential part of the ecological system may be a potential route for NP transport and bioaccumulation into the food chain [7,8,9]. Therefore, there is a great need to investigate the effect of NPs on plants including their uptake, translocation as well as their potential toxicity. It has been reported that the impact of NPs on plants can be diverse and depends on the type of NPs, their physicochemical properties (e.g., particle size, shape or surface properties), concentration, time exposure as well as plant species [10,11,12,13,14,15]. For example, Yin et al. [16] found that in *Lolium multiflorum* roots, accumulation of silver NPs (AgNPs) of 6 nm in diameter was higher than for 25 nm. Moreover, 6 nm AgNPs more strongly affected plant growth. Another study showed that AuNPs of different sizes were accumulated by tobacco but were not found to be taken up by wheat [7,17]. AgNPs at low concentration (up to 30 μg/mL) did not penetrate *Oryza sativa* roots, however, they caused an increase in root growth. AgNPs at higher concentration (60 μg/mL) passed to the cells and had a toxic effect on the roots [18]. These findings confirm that a dose and physical properties of NPs affect their availability and reactivity in plants. However, the surface chemistry of NPs is also very important as it may influence NP reactivity, penetration and movement within the plant and therefore plant responses to the same type of NPs may be completely different [19]. To date, only a few studies have demonstrated the importance of the coating properties on the NPs uptake and their effect on plants. Zhu et al. [20] have proven that the surface charge of AuNPs has an impact on diversity in their uptake by different plant species and accumulation on the root surface. Similar results have been observed on tomato and rice since (+) AuNPs (positively charged) more readily adhered to the roots and were easily internalised, while (−) AuNPs (negatively charged) were less taken up by plants [21]. Other studies revealed that the rate and extent of CdSe/CdZnS quantum dots absorption by poplar trees also depend on their surface properties [22]. One more important issue in NP-plants interaction is a cell wall which is the first physical barrier for entry of NPs from the external environment. The sieving properties of the plant cell wall impose a limitation on the size of particles that can easily pass through it. The size exclusion limit for the plant cell wall is determined by pore size which has been estimated to be between 3.3 to 6.2 nm [14,23,24]. Taking into account the very small diameter of wall pores, it can be assumed that the cell wall may be an impassable boundary for NPs [14,25]. However, some literature data showed that the cell wall permeability may change depending on the environmental conditions of plant growth [26,27]. A few reports indicate that NPs may cause enlargement of pores in a cell wall which further facilitates the entry of large NPs [28,29]. The question arises, whether the surface charge of NPs has any influence on cell wall permeability? The knowledge of NP properties, which can determine the transport and uptake across the cells, will improve our understanding of their toxicity. In present work, we evaluated interaction of 5 nm AuNPs with different surface charge (positive, negative and neutral) with *Arabidopsis thaliana* (Arabidopsis) roots. AuNPs were selected for this study because they have been demonstrated to have many benefits compared to other NMs including their biologically inert properties [20]. AuNPs are the most stable metal nanoparticles, the core material is an inert metal and is sparingly soluble in most solvents. Moreover, compare to other NPs, AuNPs do not easily release metal ions, making them relatively easy to detect [20,30]. We chose to the study *Arabidopsis thaliana* since it is a small model plant with a short life cycle which allows easy manipulation and study. We conducted our researches on the Columbia (Col-0) because this is the most commonly used ecotype within the Arabidopsis research community (The Arabidopsis Genome Initiative, 2000). The first objective of this research was to compare the effect of AuNPs with a different coating on the Arabidopsis root histology and ultrastructure. The changes in the root’s development may suggest AuNP penetration into the roots. Thus, our next goal was to determine whether a different surface charge of AuNPs affect their internalisation to roots. In these studies, the 25 μg/mL concentration was applied in order to observe the relevant phytotoxic responses as well as potential AuNP uptake. In addition, the protoplast culture of Arabidopsis leaves was also analysed to verify the hypothesis that plasma membrane is not a barrier for nanoparticles. In this experiment, we examined the impact of the surface charge of AuNPs on their passage to the cells without the wall.

## 2. Results

### 2.1. AuNPs Change the Histology of Arabidopsis thaliana Roots

The Arabidopsis root species consist of four zones: meristemastic (MZ), transition (TZ), elongation (EZ) and differentiation (DZ; Figure 1). MZ is characterized by the small cells covered by the root cap. This region includes the stem cell niche composed of the quiescent center and surrounding stem cells [31]. The daughter stem cells divide a few times and they switch to expansion, the expanding cells form the EZ. Between these two zones is small region—TZ, where cell division rates are slow and the cells begin to increase in size. DZ is formed with elongated cells from different tissues that mature into completely differentiated cells. 

AuNPs of 5 nm in diameter had different effect on root development depending on their surface charge. (+) AuNPs did not cause any significant changes in Arabidopsis root histology in comparison to the control root (Figure 1A,B). The changes in root histology were clearly visible after treatment of neutral AuNPs. The first detectable alternation in all of the treated roots was more or less developed vacuolization of rhizodermal and cortical cells from MZ, TZ and EZ (Figure 1C1–C3). Another significant change in histology was the decrease in root MZ size in the neutral AuNPs (reduction of about 45%) and (−) AuNP treated roots (reduction of about 36%) in comparison to the control (distance between the quiescent center and the TZ, TZ is determined based on the position of the first elongating cortical cell; Table 1; Figure 1A,C1–C3, red arrowheads). (−) AuNPs caused different effect on root histology comparing to the control (Figure 1D,E1–E3), and these changes were not so recurrent as in the case of neutral AuNPs treatment. The most often observed changes were alternations of root zones that have been visible as a formation of lateral roots close to the root tip (Figure 1E1,E3) or an enhancement of root hairs development close to the root tip as a consequence of shortening the MZ and DZ (Figure 1E2). Moreover, increase in radial cell dimension of cortex cell was observed under (−) AuNP treatment (Figure 1E2) since for the control the average diameter was equal to 14.2 µm (SE = 0.6, N = 30) while for treated roots was 25.8 µm (SE = 1.04; N = 10). The obtained results indicate that AuNPs caused the shortening of the MZ and the degree of this influence depends on the surface charge that AuNPs may be caused by reduction in the cell division in MZ.

In the control roots, the rhizodermal cell ultrastructure was characterised by dense cytoplasm containing numerous ribosomes and rough endoplasmic reticulum (ER), numerous active dictyosomes, mitochondria, plastids and central spherical nucleus with large nucleolus (Figure 2A–C4). The cytoplasm of cells from roots treated with (+) AuNPs and (−) AuNPs remained unchanged (Figure 2E, Figure 3A–I), however, under treatment with neutral AuNPs, the cytoplasm was less electron-dense (Figure 2H). Rough ER was of a normal appearance under (+) AuNPs conditions (Figure 2E) but in (−) AuNP treated roots, the presence of wider lumen of ER complexes prevailed in cells (Figure 3B,C,H,I). In rhizodermal cells of neutral AuNPs treated root, the dictyosomes structure was changed and few or no vesicles were found in their vicinity (Figure 2I1,I4). No differences in the structure of dictyosomes in other treated roots were observed (Figure 2F1,F4, Figure 3C). Regarding mitochondria, in the control cells they had different shapes and part of them contained numerous cristae in a dense matrix (Figure 2C3) while some lost their cristae being characterized by the electron-light matrix in the centre with few cristae on periphery (Figure 3B,C4). Under (−) AuNPs conditions, most mitochondria revealed a dense matrix and dilated, frequently ring-shaped cristae (Figure 3B–I). Other AuNPs did not cause structural changes in mitochondria structure in comparison to the control (Figure 3E,F3,H,I3,I4). The plastids remained unchanged upon treatment with neutral AuNPs and (+) AuNPs and as in the control they had electron-dense stroma with few lamellae and were characterised by the absence of thylakoids and some of them had visible plastoglobule (Figure 2B,C4,E,F4,H,H4). The plastids in the cells of roots treated with (−) AuNPs had light stroma without any lamellae and other structures inside (Figure 3C,E). No visible changes were detected in the nucleus under the influence of various AuNPs compared to the control (Figure 2C2,F2,I2, Figure 3B). The anticlinal cell walls of rhizodermis in all treated and untreated roots were thin and contained simple plasmodesmata (Figure 2B,H inset, E). Increased vacuolization was observed for all of the roots growing under AuNPs conditions and vacuoles were mostly filled with membranous/fibrillary structures (Figure 2E,G,H, Figure 3E–I). However, in the (−) AuNPs treated roots, the content of vacuoles was very diverse, some of them contained less or more electron-dense aggregations of granular precipitates located centrally or peripherally in the vacuole (Figure 3C,E–H) and the irregular membranous material was also found in these vacuoles (Figure 3G–I). The most conspicuous difference between all treated roots in comparison to the control was the width of periplasmic space in rhizodermal cells (Figure 2A,D,G). This region between the membrane and cell wall was significantly wider in roots treated with neutral AuNPs and (+) AuNPs, while in the case of (−) AuNPs it was significantly reduced (Table 1). In the periplasmic space of roots growing under AuNPs conditions numerous vesicles and membranous material were visible (Figure 2D,E inset, G,H, arrows, Figure 3A,D). This lamellar and vesicular structures seem to be paramural bodies. Few paramural bodies were also observed in the rhizodermal cells from the control (Figure 2C4, arrow). Another significant modification was an increased thickness of the outer periclinal cell wall of rhizodermis, which bordered the lateral root cap in roots treated with (−) AuNPs (Table 1, Figure 3A) and their decrease in the case of neutral AuNPs and (+) AuNPs (Table 1).

The ultrastructure of the cortical cells from the MZ of control roots was similar to rhizodermal cells (Figure 4A–C). Treatment with AuNPs revealed increased secretion since a large amount of paramural bodies with different size and shape were observed comparing to the control (Figure 4A,B,D,F,G,K,J,L, arrows). All cortical cells and rhizodermal/cortical cells were syplasmically connected as the plasmodesmata were found in the relevant walls (Figure 4A–C,F,H). Under neutral AuNP conditions, the cytoplasm was less electron-dense and there were numerous small vacuoles (Figure 4E,F) or one big vacuole that occupied centre of the cell and also few or no vesicles in the vicinity of dictyosomes (Figure 4E). Conversely, in the (−) AuNPs treated roots, dictiosomes seemed to have increased reduction of vesicles (Figure 4L) compared to the control (Figure 4B). In the (+) AuNPs and (−) AuNPs treated roots, cortical cells contained numerous vacuoles of different size which mostly contained membranous or vesicular structures of different shape (Figure 4H,I,K,L). 

This analysis pointed out that all analysed AuNPs have an influence on rhizodermal cell ultrastructure. The most apparent alterations concern the width of periplasmic space, thickness of outer periclinal cell wall of rhizodermis and increased vacuolization of cells.

### 2.2. AuNPs Did Not Enter Arabidopsis Roots

Taking into consideration modifications in root histology and ultrastructure under the AuNPs treatment, it becomes interesting to investigate whether nanoparticles penetrate inside the cells. In order to verify AuNP passage to the roots, the studies were conducted using high-resolution transmission electron microscopy (HRTEM). This tool allows thorough analyses in contrasted samples. The results showed that AuNPs did not pass through the cell wall barrier of rhizodermis or root cap cells of Arabidopsis regardless of their charge. AuNPs were retained on the surface along the root length independently of the root zones (Figure 5A–54). HRTEM analysis revealed that, irrespective of the surface charge, AuNPs tend to self-assemble into clusters and they were observed in the electron (more or less) dense substances outside the roots but in the close proximity to their surface. The largest clusters were formed by (+) AuNPs (Figure 5A1,A2,A3,A4) and were mostly observed close to the root surface along MZ for all types of nanogold particles (Figure 5A3,B3,C3). The obtained results indicate that the rhizodermal cell wall of Arabidopsis roots is an effective barrier against passage of AuNPs into cell symplasm.

### 2.3. Positively and Negatively Charged AuNPs Enter the Arabidopsis Protoplast

Since the obtained results showed that AuNPs affect root growth and development, but did not penetrate into the roots, it was decided to carry out studies on Arabidopsis protoplasts in order to verified whether the plasma membrane is also a barrier or if the wall is a molecular sieve controlling the penetration of nanoparticles into cells. In this experiment, Arabidopsis protoplasts were incubated with AuNPs of different charge in culture medium and after 1 hour samples were taken to analysis. The results revealed that both (−) AuNPs (Figure 6) and (+) AuNPs (Figure 7) were internalized by the protoplasts whereas neutral AuNPs were not found in the protoplasts. 

(−) AuNPs were abundantly present in the vicinity to the PM and probably part of them was associated with the plasmalemma (Figure 6A–D,H,J). In this region AuNPs were mostly observed in groups or clusters (from several to tens or even hundreds AuNPs in a cluster). Moreover, some AuNPs were observed in the cytosol just below the PM, showing that they can easily pass through the PM (Figure 6A,C,H). Nanogold particles were also found near the invaginating PM segments which formed vesicles (Figure 6C,D). Cell membrane engulfed AuNPs from outside the cell, drew them and formed an intracellular vesicles (endosomes). Thus, AuNPs were observed in vesicles just below PM (Figure 6E,G). This result indicates participation of endocytosis as a pathway for (−) AuNPs internalization. In the cytoplasm, AuNPs were seen not only in the vesicles but also in the multivesicular bodies (MVBs; Figure 6F). (−) AuNPs were also associated with other membranous organelle, such as endoplasmic reticulum (Figure 6G), however, some single AuNPs were not linked with any organelles, just found free in the cytosol (Figure 6G,H,J). Some MVBs and vesicles contained AuNPs fused with the tonoplast releasing their contents into vacuole (Figure 6I,K). Free AuNPs were observed inside vacuoles as single particles (Figure 6I,J) and sometimes grouped in small clusters (Figure 6H,L). Detection of particles in the cytoplasm, vesicles and vacuole suggests the path of a defence system leading to immobilization of particles in vacuole.

A similar experiment was performed using (+) AuNPs. Observation of samples showed that these particles were internalized into protoplast cells more abundantly (at least on eye estimation; Figure 7) than (−) AuNPs. Moreover, in comparison to the (−) AuNPs, positively charged nanogold were always observed in larger or smaller clusters and never as a single particle. These particles were distributed on the external surface of protoplasts, on the PM and just below PM in the cytosol (Figure 7A–C). AuNPs were also observed to bind to membranous compartments immediately below the PM (Figure 7B,C). Probably (+) AuNPs also may easily pass through PM. However, these particles were detected inside the vesicles that sometimes contained electron-dense material (Figure 7D) as well as AuNPs were bound to the membranes of vesicles (Figure 7E). These vesicles probably represented endosmoes. In some cases, the release of (+) AuNPs to cytosol from vesicles (Figure 7F) and their free presence in the cytoplasm was observed (Figure 7F,H,L). Additionally, the nanogold particles were detected in the MVBs and were inside the vesicles or associated with the membranes (Figure 7G). Observations also showed presence of AuNPs in the vacuole where they were located in the lumen in the electron dense material (Figure 7I) and sometimes were associated with the membranous structures (Figure 7J). These findings may suggest that another pathways of NPs uptake could be endocytosis. This may lead to immobilisation of AuNPs. Moreover, AuNPs were also found in the proximity to the forming extracellular vesicles composed of PM (Figure 7K,L). These vesicles were probably released from cell that may indicate that (+) AuNPs nanoparticles possibly enter to the exocytosis pathway.

## 3. Discussion

### 3.1. Changes in the Root Architecture and Histology

The root is a first plant contact with potentially toxic agents in the rhizosphere. Thus, it is the first organ that may respond to stress conditions. The changes in root system architecture upon abiotic stresses are well documented [32,33,34,35,36,37,38,39,40,41,42,43,44,45,46,47,48,49,50,51,52,53,54,55,56,57]. In terms of NP impact on root development, the best described alternation is the root growth inhibition. One of the first reports of NP toxicity on plants revealed that 13 nm aluminium oxide NPs inhibited root elongation in cabbage, carrots, cucumber, maize and soybeans [32]. Similar results after treatment with aluminium NPs were observed in *Triticum aestivum* [33]. The reduction in root length was also observed in *Lolium perenne* treated with zinc oxide NPs [34], in *Glycine max*, *Arabidopsis thaliana* and *Oryza sativa* treated with copper oxide NPs [35,36,37], in *Arabidopsis thaliana* treated with iron oxide NPs [38] and *Avena sativa* treated with silica NPs [39]. This evident symptom of NP toxicity may be correlated with reduction of MZ size since its activity controls root growth and development. However, regarding the influence of NPs, this phenomenon is poorly documented. The studies on barley and (+) AuNPs revealed that high concentration (50 µg/mL) caused ~18% decrease in MZ length while lower concentration (25 µg/mL, the same as in our research) did not affect the MZ length [40]. Our results also indicate that the MZ length of *Arabidopsis thaliana* roots is not changed upon (+) AuNPs, however, is significantly shortened under the influence of neutral AuNPs and (−) AuNPs (with an approximately 45% and 36% decrease in root MZ size, respectively). Reduction in MZ length seems to be a widespread reaction in plant response to abiotic factors such as salinity stress [41,42], cold stress [43], glucose excess [44], X-ray treatment [45] or heavy metals [46,47]. Inhibitory effect of different stressors on MZ length may be caused by either a decreased rate of cell divisions or a more rapid elongation and differentiation of cells. Yuan et al. [48] found that reduced root MZ length is a result of increased elongation-differentiation rate rather than a decrease in meristematic cell divisions. Diminished meristematic cell division potential was observed, among others, under deficiency of phosphate or excess of copper [49,50]. Disorders in root activity upon stress conditions can also cause other changes in root growth patterns, for example enhanced formation of lateral root or increased root hair density [47,51,52]. These alterations accelerate water and nutrient uptake by plants. Intensified formation of lateral root was observed under zinc oxide NP treatment of Arabidopsis [51], *Triticum aestivum* [52] or *Festuca rubra* roots [47]. The roots of wheat exposed to copper oxide NPs exhibited increase formation of root hairs close to the root tip [52]. Similar observations were noted in the case of Arabidopis exposed to copper oxide NPs [53]. In the studies presented here the changes in the root architecture were randomly observed in seedlings growing upon (−) AuNPs conditions. We found enhancement of root hairs development as well as the formation of lateral roots close to the root tip. However, there are also reports that NPs may cause hairless phenotype in roots as in the case of barley roots treated with (+) AuNPs [40]. Our results also indicate that treatment with neutral AuNPs caused strong vacuolization mainly in the rhizodermal and cortical cells of the root tip. The more vacuolated cells of the root tip have been observed upon treatment with AgNPs in the *Eruca sativa* [54] or *Lolium multiflorum* [16]. Moreover, an extension of vacuolar system seems to be a common feature of heavy metal toxicity in the root cells, for instance of Zn and Ni [47,55], Al [56] or Cd [57]. Appearance of vacuolization in plant cells may be considered as detoxification mechanism [57].

### 3.2. Changes in the Root Ultrastructure

Besides the effects of NPs on the plant development and morphology, they can also affect the plant on the ultrastructural level. However, there are insufficient data on such effects of AuNPs on plants. Most studies have shown that ultrastructural modifications under NP conditions are associated with their penetration to plant cells [39,58,59,60,61]. The alterations in root tip exhibited, among others, changes in plastids structure and appearance of protein bodies in vascular tissue [59], vacuolar changes, mitochondrial swelling and cristae degeneration [60] or plasmalemma detachment from the wall in cortical cells [61]. Our analysis has shown that regardless of the surface charge of AuNPs, they were not able to penetrate root cells and they were found in the vicinity to the root surface. However, the presence of AuNPs of different charge in growing medium has influenced root cells ultrastructure. The main changes concerned rhizodermal cells from MZ. Treatment with neutral AuNPs and (+) AuNPs caused widely detachment of PM that formed a thicker periplasmic space comparing to the control. This periplasm contained fibrillary or membranous constituents including paramural bodies. The increased space between the wall and plasmalemma suggests a process of intense secretion that occurs in the cell. This activity is often associated with direct or indirect defence against abiotic agents. Secretory products are removed from the cells through plasmalemma and are released into the periplasmic space, from which they pass through the cell wall [62]. Our results showed that in roots that had been growing with (−) AuNPs, the periplasmic space was strongly reduced but the cell wall thickness was significantly increased comparing to the control. It has been noted that the thickened cell wall may facilitate detoxification processes and confers greater mechanical resistance to cell collapse [63,64]. Moreover, cell wall thickening can act as a barrier limiting particles absorption into the protoplast like in the case of heavy metals, for instance aluminium [65], lead [66,67] or cadmium [68].

### 3.3. Internalization of Nanoparticles into the Cells

The cell wall provides a strong barrier for nanoparticle internalization into the cells. In our previous work we stated that 5 nm neutral AuNPs are not able to pass through the cell wall of barley roots since the wall pores diameter are smaller than the NP size [14]. Therefore, we tried to verify whether the different surface charge of NPs may influence the physical properties of the wall thus facilitating NP passage. The results exhibited that neutral, positively and negatively charged AuNPs at size 5 nm did not penetrate Arabidopsis roots but, as was mentioned above, had an influence on root development. However, the other studies revealed that AuNP uptake and distribution depend on both nanoparticle surface charge and plant species [20]. This research revealed that negatively charged NPs were most efficiently taken up by the plant roots while (+) NPs were extensively accumulated on the root surface [20]. In our work we also observed that (+) AuNPs most abundantly adhered to the root surface, which may be caused by electrostatic interactions between (+) AuNP and negatively charged root surface [69]. This finding is consistent with other studies focusing on the interactions of differentially charged NPs with plants [21,22,70]. In order to verify whether only cell wall is a protective barrier against NP penetration we examined differentially charged AuNPs movement into the protoplasts. We found that neutral AuNPs did not penetrate the cells while (+) and (−) AuNPs easily crossed the PM. These results are in accordance with studies on mammalian and human cell lines [71]. The neutral AuNPs are coated with a highly hydrophilic molecule—polyethylene glycol (PEG). It has been shown that increase hydrophilicity led to reduction in AuNPs uptake by HeLa cells, however, the negative charge of citrate coated AuNPs increases the AuNPs uptake in these cells as have been shown in [72]. Decreased penetration of PEG coated NMs to cells have been also shown for macrophages [73,74,75]. This effect can be associated with surfactant properties of PEG since molecules adhesion is hindered by the hydrate shell of NPs [71]. Positive surface charge caused by the presence of amino groups has been reported to increase cell surface affinity and uptake of NPs by different cell lines [76,77]. These results are consistent with our observations which showed that (+) AuNPs the most abundantly crossed the PM in protoplasts.

It has been established that in plants NPs may pass through PM *via* endocytotic or non-endocytotic pathway [78]. Endocytosis is a major process of active transport across the PM that begins with vesicle formation at the PM. Subsequently, the cargo is transported to the early and late endosomal compartments where is sorted and recycled back to the PM or directed to the vacuole for degradation [79,80]. Etexberria et al. [81] revealed that 40 nm polystyrene nano-spheres and 20 nm quantum dots were taken up by fluid phase endocytosis to distinct intracellular compartments of sycamore protoplasts. Endocytosis mechanism for NPs of similar size but a different surface charge may also be different as was shown in the experiment in tobacco protoplasts with AuNPs of opposite charge. (+) AuNPs were bound to the PM and were passed to the cells more abundantly than (−) AuNPs. The authors claim that (+) AuNPs were mostly recycled to the cell surface while (−) AuNPs were preferentially directed to the degradation pathway and that recycling to PM was slower or did not occur [82]. Similarly, the selective endocytotic pathway was found in the tobacco pollen tubes exposed to (+) AuNPs and (−) AuNPs [83]. Our results also showed that the internalization of AuNPs occurs via endocytosis and that this process differs for particles of different surface charge. Moreover, we have also observed that AuNPs may be uptaken by cells through non-endocytotic pathway as they easily passed across the PM. This is in accordance with research on *Catharanthus roseus* protoplast where multiwalled carbon nanotubes passively passed through the PM without entrapment into the degrading endosomal organelles [84].

## 4. Materials and Methods

### 4.1. Characterisation of Gold Nanoparticles

AuNPs (5 ± 2 nm) were obtained from nanoComposix Europe, Prague, Czech Republic. AuNPs surface was modified by: 1/polyethylene glycol (PEG) that neutralizes charge, 2/branched polyethyleneimine (BPEI) containing the amino groups that cause the formation of (+) AuNPs and 3/citrate that cause formation of the (−) AuNPs. All solutions of AuNPs were verified in HRTEM before starting the experiments (Figure 8A–C).

### 4.2. Arabidopsis in Hydroponic Culture

*Arabidopsis thaliana* (L.) Heynh ecotype ‘Columbia’ seeds were surface sterilized by immersion in 20% sodium hypochlorite, then kept at 4 °C in sterile water for three days for synchronized germination. The hydroponic culture was conducted as described previously [85] with some modifications. Material for growing plants contained: a liquid medium container (plastic box covered with a plate with holes) and a seed-holders (cut off the upper part of the 0.8 mL eppendorfs, 5 mm length, they fit into holes in the container). Seed-holders were filled with 0.8% agar enriched with ½ (half-strength) MS medium (Murashige and Skoog) [86] (pH = 5.8). Seeds were immersed in 0.2 % agar and placed on agar in seed-holders. The container was filled with ½ MS liquid medium and the bottom of seed holders was dipped into the nutrient solution. The seeds were cultivated for 7 days only in ½ MS medium. After 7 days, seedlings were transferred to the different types of AuNPs solutions (neutral, positively and negatively charged) at concentration 25 μg/mL (diluted in ½ MS solution) for next 7 days. The solutions of NPs were placed in the 1.5 mL eppendorfs and the holders with seedlings were placed on their lids. Then, they all were placed in the container with appropriate holes. The container was filled with water and covered with a perforated film to provide moisture. Seedlings were grown in a growth chamber at 20–23 °C, with a 16-h light/8-h dark cycle. Samples were fixed mechanically with the agarose method for obtaining straight roots [87]. First, whole seedlings were prefixed in 50 mM cacodylate buffer (Serva, Heidelberg, Germany; pH 7.0) containing 3% glutaraldehyde (Sigma-Aldrich, St. Louis, MO, USA) and 0.5% paraformaldehyde (Polysciences, Eppelheim, Germany) for two hours followed by washing three times in cacodylate buffer. Next, roots were encesed in 0.6% low melting temperature SeaPlaque agarose (Lonza, Basel, Switzerland) according to Wu et al. [87] and incubated in fresh fixative overnight at 4 °C. Roots were washed three times in cacodylate buffer and stained with 0.5% ruthenium red in cacodylate buffer for three hours. After washing in cacodylate buffer, samples were post-fixated in 1% osmium tetraoxide (Serva, Heidelberg, Germany) for 2 h at room temperature, rinsed three times in cacodylate buffer, dehydrated in graded ethanol series and gradually embedded in Epon resin (Poly/Bed 812; Polysciences, Eppelheim, Germany). Ultrathin longitudinal sections of 70 nm were cut with the use of the Leica EM UC6 ultramicrotome and collected onto carbon-coated copper grids (200 mesh, Electron Microscopy Science, Hatfield, PA, USA). Samples were stained with a uranyl acetate (Polysciences) and lead citrate agents [14] (Sigma-Aldrich, St. Louis, MO, USA) and analysed in a Jeol JEM-3010 HRTEM (300 kV) equipped with an EDS (Energy Dispersive Spectrometry, IXRF Systems Inc., Austin, TX, USA) spectrometer and a 2 k × 2 k Orius 833 SC200D CCD camera (Gatan, Pleasanton, CA, USA).

For histological analysis 1.5 µm longitudinal sections were cut and stained with Periodic Acid Schiff’s agent and counterstained with Toluidine blue (O’Brien and McCully, 1981).

### 4.3. Protoplast Culture

Sterile seeds of Arabidopsis Col-0 were placed on a medium containing ½ MS, 1% sucrose and 0.8% agar in sterile glass containers [88]. Leaves of 4- to 5-week-old sterile plants were taken for protoplast isolation. Protoplasts were prepared as previously described [89]. In brief, the sterile leaves were cut on 0.5–1 mm strips and were digested in enzyme solution [about 20 leaves for 10 mL of enzyme solution which contained: 1–1.5% cellulose R10 (Duchefa, Haarlem, the Netherlands); 0.2–0.4% macerozyme R10 (Duchefa, Haarlem, the Netherlands); 0.4 M mannitol; 20 mM KCl, 20 mM MES buffer (2-[N-morpholino]ethanesulfonic acid), pH = 5.7; 10 mM CaCl_2_]. The tissues were infiltrated in a vacuum in the dark, next digestion was continued for 3 h in the dark, without shaking at room temperature. The final step of the release of protoplasts was shaking at 80 rpm for 3 min. After cell wall digestion, protoplast suspension was filtered through a 75 µm nylon mesh to separate protoplasts from undigested tissue. Following by centrifugation (centrifuge MPW-380R) pelleted protoplasts were re-suspended by gentle shaking in 10 mL of W5 solution (washing solution; 154 mM NaCl; 125 mM CaCl_2_; 5 mM KCl2; 2 mM MES, pH 5.7). Washing was repeated two more times. Freshly isolated protoplast were re-suspended in culture medium [B5-Gamborg’s solution, 0.4 M glucose, and 1 mg/L 2,4-dichlorophenoxyacetic acid, 0.5 mg/L 6-benzylaminopurine [88]] enriched with different types of AuNPs at concentration 10 μg/mL (the dose of AuNPs has been selected based on the viability of protoplast that were subjected to different NPs concentrations). The incubation of protoplasts with nanoparticles and the control without addition of AuNPs lasted for 1 h at 26 °C in the dark (about 1 × 10^5^ protoplasts per sample). After washing protoplasts two times in the culture medium, they were embedded in 1.2% SeaPlaque agarose as was described previously [90]. After agarose solidification, samples were fixed overnight at 4 °C in 2% glutaraldehyde in the culture medium (pH 6.8), postfixed with osmium tetroxide for 3 h and next dehydrated, infiltrated and subsequently embedded in Epon resin. For ultrastructural analysis ultrathin sections, 70 nm thick, were cut with the use of Leica EM UC6 ultramicrotome and placed on grids with carbon film (200 mesh). Next, sections were stained with uranyl acetate and lead citrate agents. The samples were analysed in a HRTEM Jeol JEM-3010 (300 kV) equipped with EDS spectrometer and Gatan 2k × 2k Orius TM 833 SC200D CCD camera. 

Cell viability was assessed by staining protoplasts with fluorescein diacetate (FDA; Sigma-Aldrich, St. Louis, MO, USA) before fixation according to Anthony et al. [91] and Skálová et al. [92] (Figure 9). A stock solution that was prepared in acetone (5 mg/mL) was dissolved in culture medium (0.5 mL per 24.5 mL) in order to obtain the FDA working solution. The viability of freshly isolated protoplast was approximately 69% however after one hour of treatment, it was approximately 60%, 53% and 48% for (−) AuNPs, neutral AuNPs and (+) AuNPs, respectively. Samples were observed by Nikon Eclipse Ni-U fluorescence microscope equipped with a Nikon Digital DS-Fi1-U3 camera with corresponding software (Nikon, Tokyo, Japan).

### 4.4. Data Analysis

Measurements of the meristematic zone length, cell diameter, thickness of the cell wall and width of periplasmic space were carried out using ImageJ software (version 1.49; http://imagej.nih.gov/). For the measurements of the MZ length, the border between the quiescent centre and the TZ (indicating the position of the first elongated cortex cell) was measured as described previously [93]. At least 10 roots were subjected to this analysis. The statistical Student’s *t*-test was applied to compare the radial diameter of the cortical cells. It was measured from five cells above MZ (for AuNPs treated roots only root from Figure 3E2 was subjected to the analysis, for the control cells from three roots were measured). The periplasmic space width and cell wall thickness were measured along the MZ from at least two roots. Differences between means were compared using Dunnett’s test implemented in the Statistica v.12 software.

## 5. Conclusions

Taking together the obtained results we can conclude that:

(1) A different surface charge of AuNPs influence Arabidopsis root development, which has been confirmed on the histological and ultrastructural level.

(2) The cell wall of Arabidopsis rhizodermis is an impermeable barrier for AuNP penetration regardless of the surface charge of particles.

(3) Positively and negatively charged AuNPs enter through the plasma membrane, which was confirmed by the experiment on protoplast culture. The results suggest that the pathway of a defence system that leads to immobilization of AuNPs varies depending on the surface charge.

## Figures and Tables

**Figure 1 ijms-20-01650-f001:**
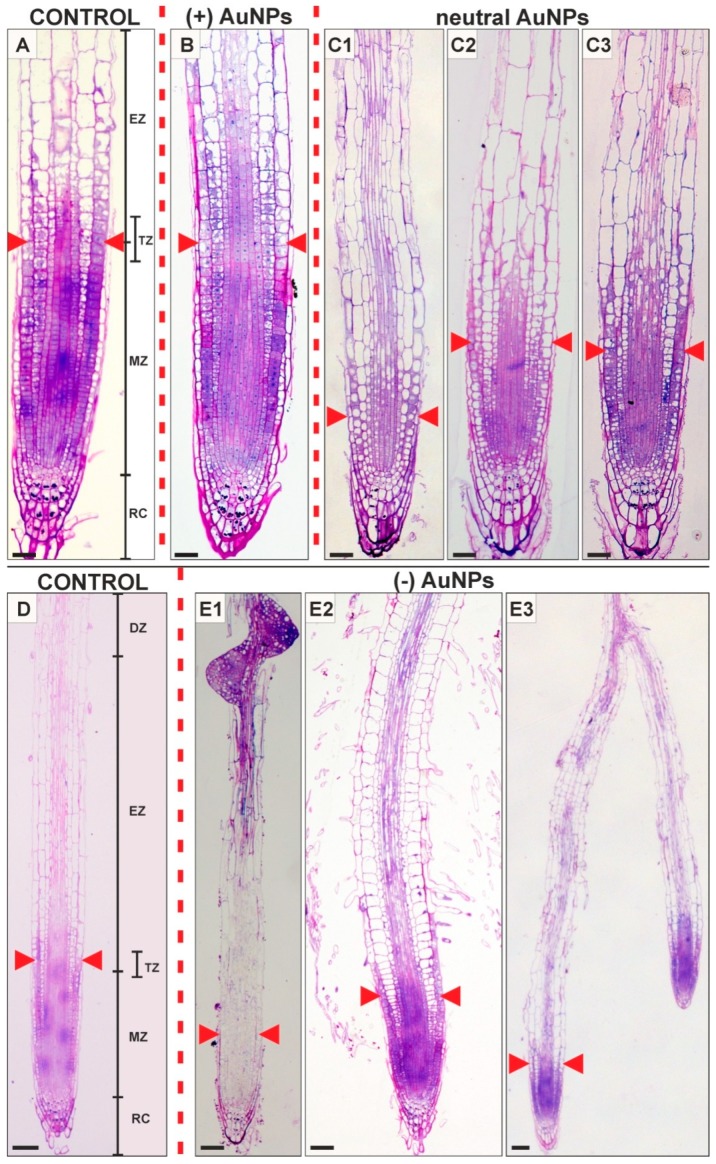
Longitudinal sections of *Arabidopsis thaliana* roots of seedlings that have been growing in control conditions (**A**,**D**) and treated with: (−) AuNPs (**B**), neutral AuNPs (**C1**–**C3**) and (+) AuNPs (**E1**–**E3**). Red arrowheads indicate meristem boundary (the position of the first elongating cortical cell); DF—differentiation zone; EZ—elongation zone; MZ—meristematic zone; RC—root cap; TZ—transition zone. Scale bars: **A**–**C2** = 20 µm; **D**–**E3** = 50 µm.

**Figure 2 ijms-20-01650-f002:**
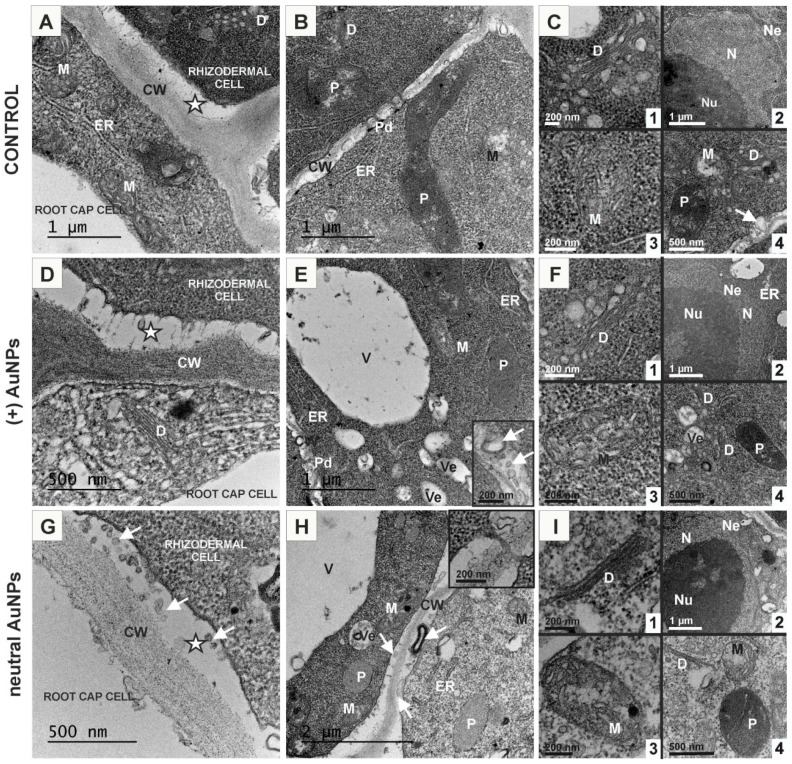
Ultrastructure of rhizodermal cells from the control roots (**A**–**C**); treated with (+) AuNPs (**D**–**F**) and neutral AuNPs (**G**–**I**). CW—cell wall, D—dictyosome, ER—endoplasmic reticulum, M—mitochondria, N—nucleus; Ne—nuclear envelope; Nu—nucleolus; P—plastid; Pd—plasmodesmata; V—vacuole; Ve—vesicle; arrows—paramural bodies; asterisks—periplasmic spaces.

**Figure 3 ijms-20-01650-f003:**
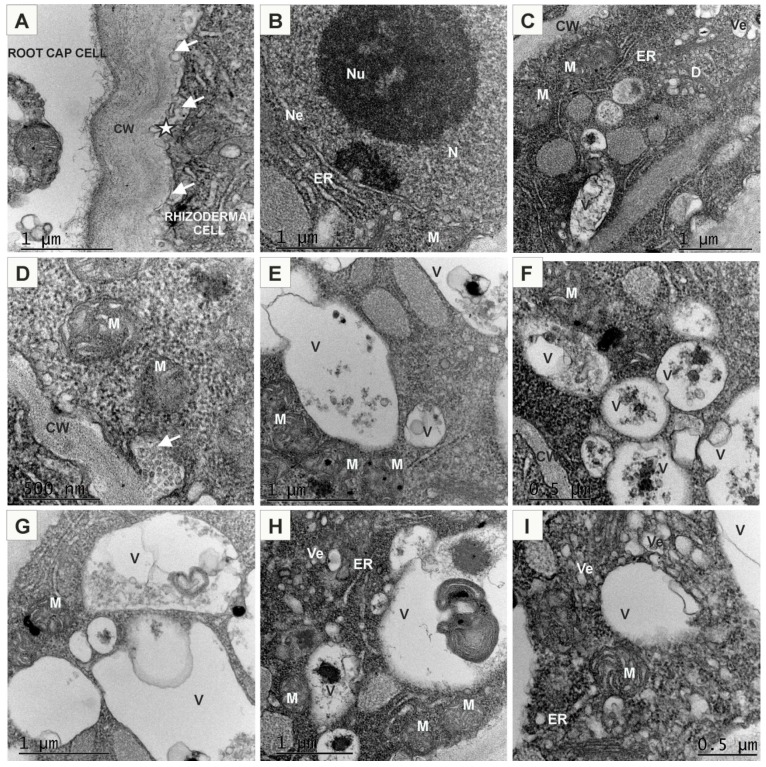
Ultrastructure of rhizodermal cells from the roots treated with (−) AuNPs (**A**–**I**). CW—cell wall; D—dictyosome; ER—endoplasmic reticulum; L—leucoplast; M—mitochondria; N—nucleus; Ne—nuclear envelope; Nu—nucleolus; Pd—plasmodesmata; V—vacuole; Ve—vesicle; arrows—paramural bodies; asterisk—periplasmic space.

**Figure 4 ijms-20-01650-f004:**
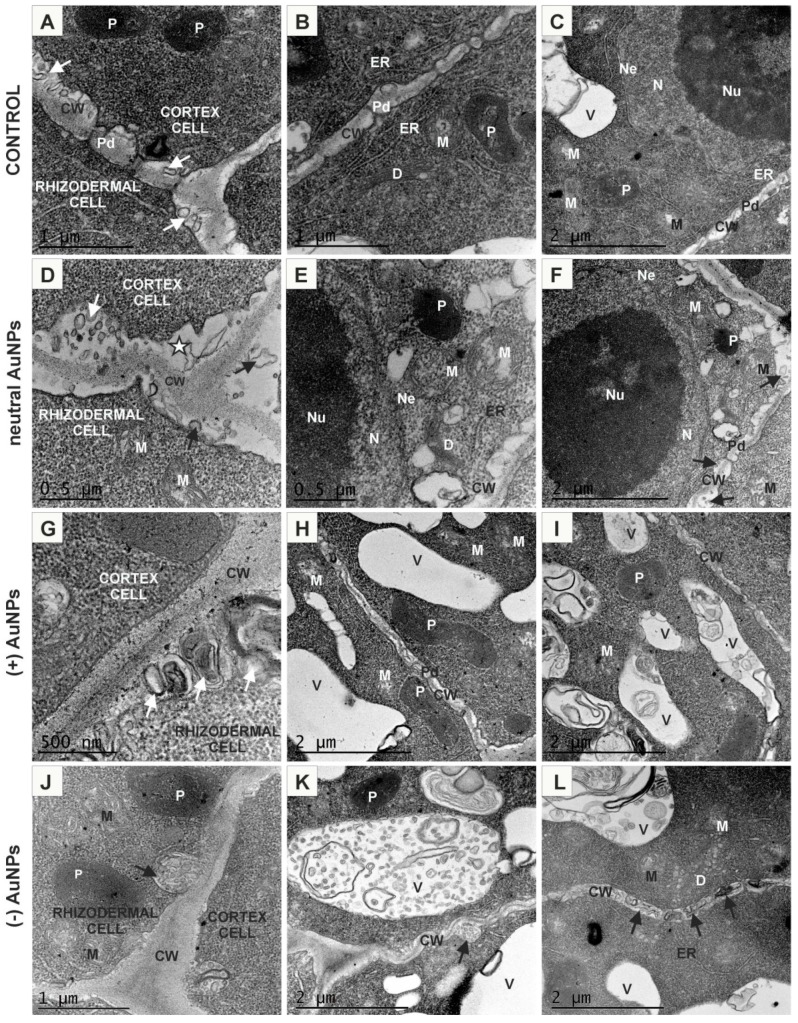
Ultrastructure of cortical cells from the control roots (**A**–**C**); treated with neutral AuNPs (**D**–**F**), (+) AuNPs (**G**–**I**) and (−) AuNPs (**J**–**L**). CW—cell wall; D—dictyosome; ER—endoplasmic reticulum; M—mitochondria; N—nucleus; Ne—nuclear envelope; Nu—nucleolus; P—plastid; Pd—plasmodesmata; V—vacuole; Ve—vesicle; arrows—paramural bodies; asterisk—periplasmic space.

**Figure 5 ijms-20-01650-f005:**
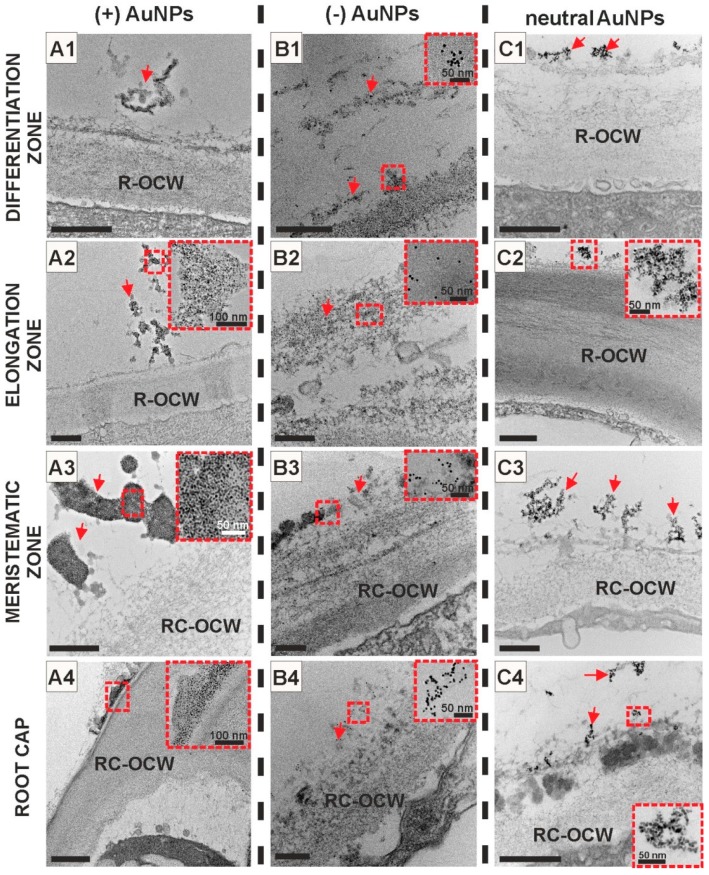
Ultrastructure of *Arabidopis thaliana* root surface along all of root zones. AuNPs regardless of their surface charge: (+) AuNPs (**A1**–**A4**), (−) AuNPs (**B1**–**B4**) and neutral AuNPs (**C1**–**C4**) did not pass the cell wall and they were retained on the root surface; red arrows indicate AuNPs, insets—high resolution images of AuNPs. R-OCW—root outer cell wall, RC-OCW—root cap outer cell wall. Scale bars: **A1**–**A4**, **C1**–**C4** = 500 nm; **B1**–**B4** = 200 nm.

**Figure 6 ijms-20-01650-f006:**
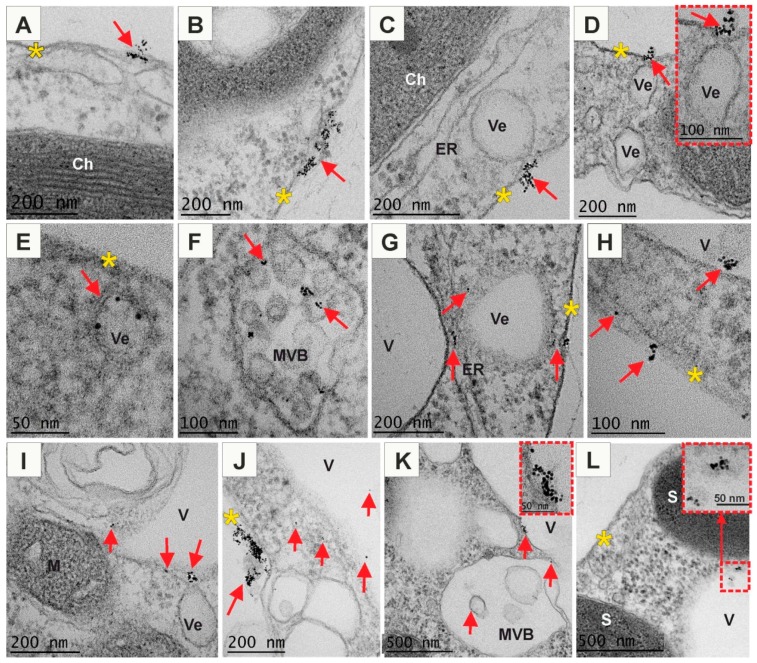
Ultrastructure of Arabidopsis protoplasts treated with (−) AuNPs (**A**–**L**). (−) AuNPs passed through the PM that provided the first barrier between the environment and cytoplasm. (−) AuNPs were detected on the PM surface, bound to the PM or in cytoplasm just below PM (**A**,**B**,**H**). They were also mostly found in the vesicles or in the vicinity of the vesicles (**C**,**D**,**E**,**G**), in the multivesicular bodies (**I**), in the cytoplasm (**G**,**H**,**J**) or in the vacuole (**H**,**V**,**J**,**K**,**L**); red arrows indicate the presence of AuNPs; yellow asterisks indicate plasma membrane barrier; Ch—chloroplast; ER—endoplasmic reticulum; M—mitochondria; MVB—multivesicular body; S—starch grain; V—vacuole; Ve—vesicle.

**Figure 7 ijms-20-01650-f007:**
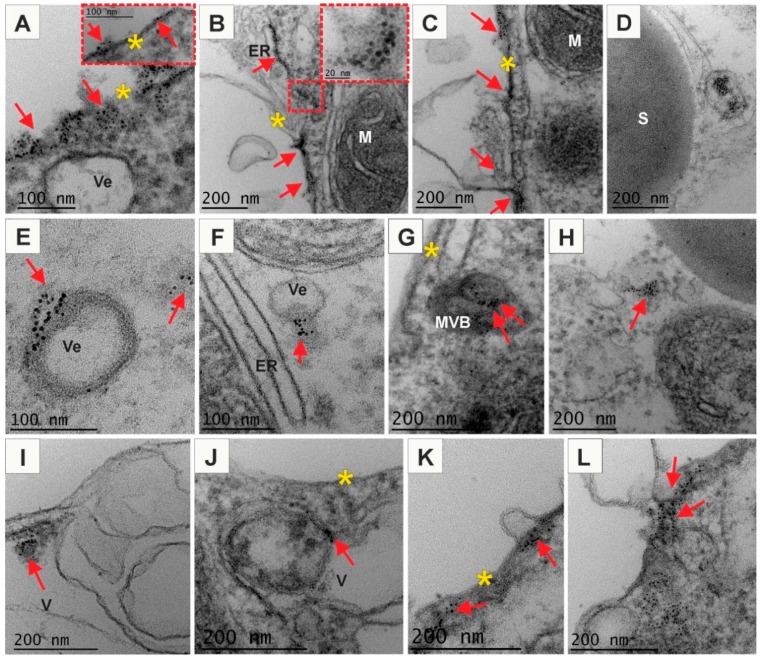
Ultrastructure of Arabidopsis protoplasts treated with (+) AuNPs (**A**–**L**). (+) AuNPs passed through the PM that provided the first barrier between the environment and cytoplasm. (+) AuNPs were found on the outer and inner side of PM and associated to PM (**A**–**C**). They were also observed in the membranous compartment, among others in endoplasmic reticulum (**B**,**C**); in the vesicles or in the vicinity of the vesicles (**D**–**F**,**J**), in the multivesicular bodies (**G**,**I**), in the cytoplasmic compartments (**E**,**H**,**K**,**L**) or in the vacuole (**I**,**J**); red arrows point to AuNPs; yellow asterisks indicate plasma membrane; ER—endoplasmic reticulum; M—mitochondria; MVB—multivesicular body; S—starch grain; V—vacuole; Ve—vesicle.

**Figure 8 ijms-20-01650-f008:**
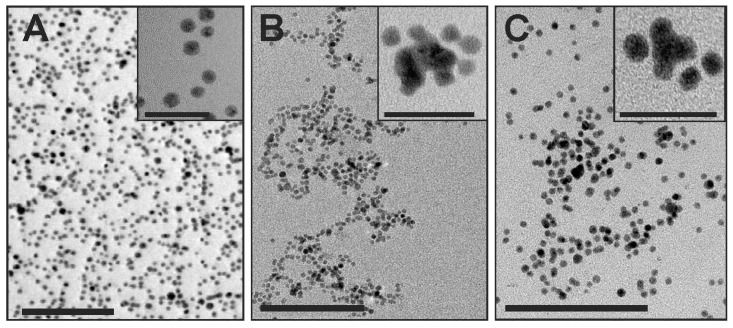
High-resolution transmission electron micrographs of different 5 nm AuNPs: (**A**) neutral, (**B**) negatively and (**C**) positively charged nanogold. Insets indicate high magnification of selected AuNPs. Scale bars: **A**–**C** = 100 nm; insets **A**–**C** = 20 nm.

**Figure 9 ijms-20-01650-f009:**
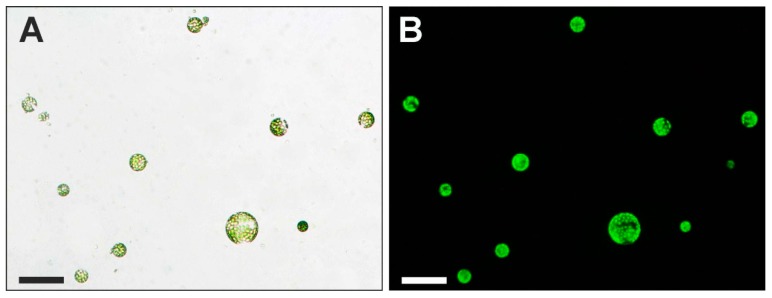
Freshly isolated protoplasts: (**A**) bright field and (**B**) stained with FDA—test of protoplast viability (UV light). Scale bars: 100 µm.

**Table 1 ijms-20-01650-t001:** The effect of AuNPs on the roots: length of MZ, width of periplasmic space and thickness of the cell wall compare to the control roots.

Characteristic	Control		Neutral AuNPs			(+) AuNPs			(−) AuNPs		
	Mean	SE	Mean	SE	*p*	Mean	SE	*p*	Mean	SE	*p*
MZ length [µm]	222.6	8.94	120.3	8.59	<0.001	227	9.34	0.974	142.2	9.34	<0.001
Periplasmic space width * [nm]	114	8.68	233.7	8.3	<0.001	173.6	8.18	<0.001	52.1	7.99	<0.001
Cell wall thickness ** [nm]	466.4	12.79	244.1	12.59	<0.001	274.2	12.36	<0.001	1103.1	12.18	<0.001

SE—standard error, *p*—significance level—the differences between means were compared using the ANOVA—Dunnett’s test, *p* value < 0.001 indicate statistically significant differences. * Width of the periplasmic space in the rhizodermal cell next to outer rhizodermal cell wall, ** thickness of the outer cell wall of rhizodermis from MZ.

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
