# Peer review of "Effect of Nanoparticles Surface Charge on the Arabidopsis thaliana (L.) Roots Development and Their Movement into the Root Cells and Protoplasts"

_ijms, 2019, doi:10.3390/ijms20071650_

Round 1

Reviewer 1 Report

Very nice work with many results, however manuscript is basically divided in two parts that are not fully connected. But it is only minor comment.

I miss data from protoplasts viability. How big was protoplast viability before fixation? How did differ viability between freshly isolated protoplasts and protoplasts in culture media enriched with all studied AuNPs and protoplasts in culture medium without any AuNPs? Were detected charged AuNPs in viable protoplasts or death ones?

Please, add data concerning protoplasts viability.

Author Response

Dear Reviewer,

We appreciate very much the Reviewer for the constructive comments and for the effort and time that was putted into the revision process of the manuscript. We are very grateful to the Reviewer for their comments and thoughtful suggestions. We believe that the Reviewer comments have strengthened our manuscript. Based on these comments and suggestions, we have made careful modifications to the manuscript.

Manuscript number: ijms-470606

Title: Effect of nanoparticles surface charge on the Arabidopsis thaliana (L.) roots development and their movement into the root cells and protoplasts

Answers to Reviewer comments

1.      Very nice work with many results, however manuscript is basically divided in two parts that are not fully connected.

We are very grateful for the appreciation of our work. We do agree that obtained results make the work divided into two parts. We have changed the order of the subsections from the results section and as first we describe histological and ultrastructural changes under AuNPs treatment and next penetration of NPs to the roots and protoplasts. To combine these two parts we modified first sentence of section regarding the AuNPs penetration to the roots:

Taking into consideration modifications in root histology and ultrastructure under the AuNPs treatment, it becomes interesting to investigate whether nanoparticles penetrate inside the cells.”

We also have introduced a sentence in the beginning of the protoplast section:

Since the obtained results showed that AuNPs affect root growth and development, but did not penetrate into the roots, it was decided to carry out studies on Arabidopsis protoplasts in order to verified whether the plasma membrane is also a barrier or if the wall is a molecular sieve controlling the penetration of nanoparticles into cells.”

We do hope that these changes makes work better in reception.

2.      I miss data from protoplasts viability. How big was protoplast viability before fixation? How did differ viability between freshly isolated protoplasts and protoplasts in culture media enriched with all studied AuNPs and protoplasts in culture medium without any AuNPs? Were detected charged AuNPs in viable protoplasts or death ones?  Please, add data concerning protoplasts viability.

We appreciate your comments very much. As you recommended, we have performed the additional data in the material and method section:

“Cell viability was assessed by staining protoplasts with fluorescein diacetate (FDA; Sigma-Aldrich) before fixation according to Anthony et al. [91] and Skálová et al. [92] (Figure 9). (...) The viability of freshly isolated protoplast was approximately 69% however after one hour of treatment, it was approximately 60%, 53% and 48% for (-) AuNPs, neutral AuNPs and (+) AuNPs, respectively.”

We are grateful for your work put into the improvement of the manuscript.

On behalf of the Authors,

Anna Milewska-Hendel

Author Response

Dear Reviewer,

We appreciate very much the Reviewer for the constructive comments and for the effort and time that was putted into the revision process of the manuscript. We are very grateful to the Reviewer for their comments and thoughtful suggestions. We believe that the Reviewer comments have strengthened our manuscript. Based on these comments and suggestions, we have made careful modifications to the manuscript.

Manuscript number: ijms-470606

Title: Effect of nanoparticles surface charge on the Arabidopsis thaliana (L.) roots development and their movement into the root cells and protoplasts

Answers to Reviewer comments

1.      In this manuscript, the authors examined the permeation of positively charged, negatively charged and neutral gold nanoparticles through cell walls of plant roots. However, their results seem to indicate that these gold nanoparticles did not cross the cell wall regardless of the surface charge of the gold nanoparticles. Nonetheless, these nanoparticles have caused diverse changes in root histology and ultrastructure. While the reported work appeared to be interesting, it is not easy to follow the description and to understand the discussion because the manuscript has not been prepared for non-experts in the area. Also, the separation of results and discussion into two sections has also made the manuscript excessively long to read.

We are very grateful for the appreciation of our work. We are aware that the work is extensive and described in detail however, we suppose that without this, it would be difficult to draw conclusions. For better reading we have changed the order of the subsections from the results section and as first we describe histological and ultrastructural changes under AuNPs treatment and next penetration of NPs to the roots and protoplasts. Now it is connected with the order in the discussion section. To combine these two parts we modified first sentence of section regarding the AuNPs penetration to the roots:

Taking into consideration modifications in root histology and ultrastructure under the AuNPs treatment, it becomes interesting to investigate whether nanoparticles penetrate inside the cells.”

We also have introduced a sentence in the beginning of the protoplast section:

Since the obtained results showed that AuNPs affect root growth and development, but did not penetrate into the roots, it was decided to carry out studies on Arabidopsis protoplasts in order to verified whether the plasma membrane is also a barrier or if the wall is a molecular sieve controlling the penetration of nanoparticles into cells.”

Separating the discussion with the results we were guided by the requirements of the journal which suggest such a type of paper. However, we believe that after our correction - ordering subsections of results and discussions, our work is better in reception.

2.      Specific comments / queries and recommended changes / corrections: 1-78

We extremely appreciate your careful reading of the manuscript and detailed comments and corrections. We've made corrections according to your suggestions. Below we answer your questions

A.    Line 73, “many benefits”, such as?

The examples of benefits of gold nanoparticles are presented in two subsequent sentences in the text:

“AuNPs were selected for this study because they have been demonstrated to have many benefits compared to other NMs including their biologically inert properties [20]. AuNPs are the most stable metal nanoparticles, the core material is an inert metal and is sparingly soluble in most solvents. Moreover, compare to other NPs, AuNPs do not easily release metal ions, making them relatively easy to detect [20, 30].”

B.     Line 75, “Moreover, compared to other NPs, AuNPs do not easily release metal ions, making them relatively easy to detect …”. If so, I do not understand why “gold nanoparticles are easy to detect because they do not easily release metal ions”?

By this statements Authors understand that the Au nanoparticles are chemically stable and do not dissolve or release ions to the cell tissue. Due to the fact that the nanopraticles are stable it is possible to observe them using TEM. Otherwise the studied material could be distributed in the cell and do not provide sufficient contrast under TEM. The other reason that the Au particles were selected is that the Au metal is sufficiently heavy giving good contrast in the TEM. On the other hand if the metal ions were released from the nanoparticles to the cell it would make it hard to detect them since they would differ in size, structure and they would have different properties compared to the initial NPs. Moreover they can react with some organic or inorganic components in the plant body and thus they can be unnoticed under HRTEM observations.

C.     Line 78, what is “Col-0”?

Col-O (Columbia) it is wild type ecotype of Arabidposis thaliana Halleri (L.). It is commonly used wild type in wide range of researches. We explained the abbreviation in the text.

D.    Line 403, “… 0.8 mL Eppendorfs, … with ½ Murashige and Skoog (MS) medium (pH 5.8) [86].” What is the unit for the quantity “½” here?

E.     Line 407, is there a difference between “MS medium” here and “½ MS medium” earlier?

F.      Line 431, so “½ MS” means “half-strength MS”! But, how would readers be able to identify this from “½ MS medium”? Wouldn’t it be more appropriate to specify the concentration?

“Half-strength MS” is commonly used nomenclature for concentration of MS medium reduced to half in in vitro cultures. Thank you for your carefully reading the work, now we introduced explanation (half-strength MS) in the first time when “½ MS” was used in the text and we corrected “MS medium” on “½ MS medium”.

G.    Line 100, the authors have simply stated here that, “… did not cause any significant changes … (Figure 1A, B)”, without initially describing the results in Figure 1A and Figure 1B and then pointing out the similarity/difference between the two figures!

In this sentence we state that there is no changes between control roots and treated with positively charged (+) AuNPs. In the following sentences we indicate the differences that were observed between negatively charged (-) AuNPs and neutral AuNPs compare to the control.

H.    Line 113, what is “SE”? What is the dimensionless quantity of “0.6”?

Thank you for your carefully reading the work. SE is standard error instead standard deviation. We are sorry for the mistake.

We are grateful for your work put into the improvement of the manuscript.

On behalf of the Authors,

Anna Milewska-Hendel
